# An Evaluation of the Impact of Barcode Patient and Medication Scanning on Nursing Workflow at a UK Teaching Hospital

**DOI:** 10.3390/pharmacy8030148

**Published:** 2020-08-19

**Authors:** Sara Barakat, Bryony Dean Franklin

**Affiliations:** 1Department of Practice and Policy, UCL School of Pharmacy, University College London, London WC1N 1AX, UK; saramiriambarakat@gmail.com; 2Centre for Medication Safety and Service Quality, Pharmacy Department, Imperial College Healthcare NHS Trust, London W6 8RF, UK

**Keywords:** barcode medication administration (BCMA), nurses’ workflow, inpatient setting, drug round

## Abstract

Barcode medication administration (BCMA) is advocated as a technology that reduces medication errors relating to incorrect patient identity, drug or dose. Little is known, however, about the impact it has on nursing workflow. Our aim was to investigate the impact of BCMA on nursing activity and workflow. A comparative study was conducted on two similar surgical wards within an acute UK hospital. We observed nurses during drug rounds on a non-BCMA ward and a BCMA ward. Data were collected on drug round duration, timeliness of medication administration, patient identification, medication verification and general workflow patterns. BCMA appears not to alter drug round duration, although it may reduce the administration time per dose. Workflow was more streamlined, with less use of the medicines room. The rate of patient identification increased from 74% (of 47) patients to 100% (of 43), with 95% of 255 scannable medication doses verified using the system. This study suggests that BCMA does not affect drug round duration; further research is required to determine the impact it has on timeliness of medication administration. There was reduced variability in the medication administration workflow of nurses, along with an increased patient identification rate and high medication scan rate, representing potential benefits to patient safety.

## 1. Introduction

Barcode medication administration (BCMA) involves scanning a patient’s unique barcode and medication barcodes to verify that these are correct before proceeding to administer a dose. It is becoming more common in many countries, often integrated with electronic prescribing and medication administration (ePMA) systems [1]. BCMA is reported as having a number of patient safety benefits, including a reduction in both the rate and severity of medication administration errors [2,3,4,5]. One UK study also found that BCMA dramatically increased the percentage of patients for whom identity was checked [6]. An international systematic review concluded that BCMA technology has potential to reduce error rates, although human factors remain that need to be addressed to fully reap the benefits [7]. Specifically, it has been proposed that if BCMA systems hinder nurses’ workflow, workarounds occur that may reduce effectiveness in practice; these include actively bypassing key steps, such as placing a patient’s barcode on another object for easier scanning [8,9]. However, little is known about the impact of BCMA on nursing activity and workflow. Results from direct observation studies suggest the time nurses spend on medication administration tasks can remain the same or reduce, and the time spent on direct care can remain the same or increase [10,11,12]. However, these studies have compared BCMA (with associated ePMA) to paper-based medication administration, whereas many hospitals implementing BCMA will already have ePMA in place [1]. Additionally, research exploring nurses’ perceptions of BCMA presents conflicting findings. Time spent on medication administration is sometimes perceived to be higher, with some nurses believing that BMCA reduces time spent on direct patient care [13,14], although it has also been argued that this extra time is helpful for ensuring correct medication verification [15]. Interviewed nurses frequently reported BCMA slowing down their workflow, partly due to operational difficulties with the technology [16]. Such conflicting findings suggest that more research is needed in this area and that the context may also be important. Our aim was therefore to evaluate the impact of BCMA, when added to an existing ePMA system, on nursing workflow in a UK hospital. Objectives were to compare the following between matched BCMA and non-BCMA wards:Duration of drug administration rounds;Timeliness of medication dose administration;Active identification of patients by nurses;Active verification of medication by nurses (BCMA only);Walking patterns of nurses on the ward.

## 2. Materials and Methods 

### 2.1. Setting 

The study was undertaken on two surgical wards at a large acute London hospital. Data were collected on a ward that had not yet implemented BCMA (a gastrointestinal surgery ward), and on a ward with the same physical layout (a vascular surgery ward) four days after BCMA implementation. Each ward had a total of 24 beds, distributed among four bays plus four side rooms. 

During each drug round, nurses administered doses from ward stock, non-stock medication dispensed to individual patients, and patients’ own medication from home. Patients’ own and individually dispensed medication was typically stored in bedside lockers; ward stock medication was stored in the medication room and/or in lockable cabinets underneath computers on wheels (COWs). About half of the ten COWs had cabinets for medication storage on the non-BCMA ward, though these were used by all staff and sometimes were not available for use during drug rounds. On the BCMA ward, five COWs were fitted with a larger medicine cabinet and designated for drug rounds only, while five COWs without cabinets remained in place for general use. Nurses accessed the ePMA system on the COWs to determine the doses due and record their administration. Prior to BCMA introduction, each individual dose had to be electronically signed to indicate that it had been given. Drug rounds were scheduled for 8am, 12pm, 2pm, 6pm and 10pm, with most medication scheduled to be given at these times. Each nurse would be responsible for one bay of five beds and possibly a side room.

The ePMA and BCMA systems were part of Cerner Millennium [17]. Following implementation, the nurse would use a barcode scanner tethered to the designated COW during drug rounds. Once the nurse navigated to a patient’s medication administration record, they would be prompted to scan the patient’s wristband to confirm their identity. The full list of that patient’s medication would then be displayed and, once each medication barcode was scanned, the system would validate these doses. The nurse could then sign electronically to indicate that all doses for that patient had been administered, rather than signing for each individually. If either the patient or medication barcode did not match the correct details, a warning message would appear and the nurse could take appropriate action to correct an error or give a reason for overriding the alert. 

### 2.2. Study Design and Data Collection

This was a comparative study with direct observation used to collect data during 8am drug rounds. Data were collected for ten consecutive weekdays during November 2019 on the non-BCMA ward, and then for ten weekdays during December 2019 on the BCMA ward. Random sampling was used before commencing observations to determine that bay “H” would be selected for observation every day. Nurses responsible for that bay at each drug round were approached and invited to give consent for observation. A pharmacy student observed the nurse concerned, aiming to be as unobtrusive as possible. The study was approved locally as a service evaluation; NHS ethics approval was not required. Data were collected on the outcome measures in Table 1:

Additionally, on eight drug rounds on each ward, a spaghetti diagram [18,19] was constructed detailing a floorplan of the ward along with the walking pattern of the nurse around the ward, from the time the drug round began to the time it ended. This was done for eight of ten days, as the first two days on each ward were spent creating the floorplan [18,19]. 

Quantitative data were entered into an Excel spreadsheet. Following visual inspection of the data to rule out any obvious skew, mean drug round duration (our a priori primary outcome measure), was calculated for each ward; an unpaired *t*-test was used to determine whether any differences were statistically significant. This was also done for mean time per patient and mean time per dose as secondary analyses. Timeliness of medication administration was determined by calculating the mean time difference between scheduled and administered dose times on each ward. The percentage of patients whose identification was checked was calculated on each ward, and a chi-square test used to identify whether any difference was statistically significant. Additionally, on the BCMA ward, the percentage of medication doses that were verified was calculated. All spaghetti diagrams were visually compared to identify differences in walking patterns and task activities.

## 3. Results

Ten drug rounds were observed on each ward; between three and six patients received one or more doses of medication on each ward. The mean number of doses administered was 17 on the non-BCMA ward (range 7 to 25) and 29 on the BCMA ward (range 21 to 38). Seven different nurses were observed on the non-BCMA ward, and eight on the BCMA ward.

### 3.1. Drug Administration Round Duration 

The mean drug round duration was 67.8 min (range 27 to 98 min) on the non-BCMA ward, and 68.0 min (range 48 to 99 min) on the BCMA ward (*p* = 0.98; *t*-test). There was wide variation in drug round duration (Figure 1), particularly on the non-BCMA ward.

The mean time per patient was 14.1 min (range 9.0 to 19.6 min) on the non-BCMA ward and 16.0 min (range 11.6 to 20.0 min) on the BCMA ward (*p* = 0.17; *t*-test). The mean time per dose was 4.2 min (range 3.3 to 5.5 min) on the non-BCMA ward and 2.3 min (range 1.8 to 3.2 min) on the BCMA ward (*p* < 0.001; *t*-test).

### 3.2. Timeliness of Medication Administration

On the non-BCMA ward, a total of 185 medication doses were due to be given, of which 165 (89%) were administered. On the BCMA ward, 363 medication doses in total were due to be given, of which 294 (81%) were administered. Doses were not administered due to patient refusal, the dose requiring review or if the dose was not available on the ward. The mean time difference between the scheduled and administered dose times was 60.3 min on the non-BCMA ward, and 67.5 min on the BCMA ward. Medication administration appeared to be more timely on the non-BCMA ward (*p* = 0.007; *t*-test). Despite drug rounds being scheduled to start at 8am, we noted that drug rounds on the BCMA ward generally started later, with seven of ten drug rounds beginning after 08:30 am, compared to the non-BCMA ward where only one began after 08:30 am.

### 3.3. Active Positive Patient Identification 

Nurses administered medication to 47 patients on the non-BCMA ward of whom 35 (74%) had their identification checked prior to medication administration. On the BCMA ward, nurses administered medication to 43 patients, of whom all 43 (100%) had their identification checked, showing an increase in positive patient identification with BCMA (*p* = 0.001; chi-square test). On the non-BCMA ward, all positive identifications involved the nurse manually checking the patient’s wristband. On the BCMA ward, BCMA was used to check patient identity in 40 cases (93%) with nurses also confirming this manually in 13 cases (30%). Nurses were not permitted to take the COW into side rooms due to infection control policy, so in the three cases (7%) where medication was administered to a side room patient, they scanned a barcode sticker from the patient’s notes instead of the wristband. In all three cases, the patient’s wristband was also checked manually.

### 3.4. Active Positive Medication Verification 

On the BCMA ward, the majority (255 of 294; 87%), of doses administered had a scannable barcode. Those without a scannable barcode included patients’ own drugs that had been repackaged in non-original packs, and insulin pens stored individually without their box. Although all other medication had scannable barcodes, some products such as nutritional supplements and vitamins could not be identified by the BCMA system, necessitating manual signing for administration. Of the 294 doses administered, 243 (83%) were scanned prior to administration. This represented 95% of scannable doses, with only 12 potentially scannable doses not being scanned. Of these 12 doses, 11 were subcutaneous enoxaparin doses prepared in the medicines room. While the boxes could have been scanned, four of the five nurses who administered enoxaparin chose to leave the box in the medicines room.

Overall, 14 cases of a BCMA system warning were observed, representing 6% of doses scanned. One (0.4% of doses scanned) was an error in which the nurse had selected the wrong formulation, selecting co-amoxiclav tablets when injection was prescribed; the nurse rectified this by selecting the correct form. The remaining 13 comprised seven cases when the dose could not be identified from the barcode, three where a variable dose had been prescribed requiring manual entry of the exact dose given, two where half a tablet was needed for the dose and the system required manual amendment as it had detected an apparent overdose, and one where the dose had been detected as being administered too early that the nurse overrode. In each case, the nurse took corrective action to either manually amend the record or enter a reason for overriding the system.

### 3.5. Observation of Nurse Walking Pattern around the Ward and General Activity 

An example of the spaghetti diagram is provided for both the non-BCMA (Figure 2) and BCMA (Figure 3) wards. Figure 2 depicts a longer drug round where most walking took place between the patients’ beds and the medicines room. This was similar for three further spaghetti diagrams, while another four had less walking around the ward. On the BCMA ward, for all eight spaghetti diagrams, there was consistently less walking by nurses to certain areas of the ward such as the medicines room, and most walking took place between patients’ beds as seen in Figure 3.

## 4. Discussion

### 4.1. Key Findings

Our results suggest that introduction of BCMA does not affect drug round duration, although comparison is challenging due to more doses per patient being given on the BCMA ward. The reduced time per dose on the BCMA ward suggests that nurses were able to administer more doses in a similar time-frame. While the findings suggest that BCMA decreased the timeliness of medication administration, this may be because drug rounds generally began later on the BCMA ward. If they had begun at approximately the same time, it is likely that no such difference would have been observed. In fact, the lower time per dose with BCMA indicates there may be potential to improve timeliness since BCMA has streamlined and harmonised processes, thus reducing the amount of walking to the medicines room to gather and prepare doses. 

Our data suggest that BCMA has led to a 100% patient identification check, since scanning of a patient barcode is required to administer medication. However, the workaround of scanning a barcode from the patient’s notes instead of their wristband for side room patients was performed by three different nurses, suggesting that this may have become relatively common practice. This is potentially counterproductive as it could increase the risk of errors relating to the wrong patient.

The high scan rate of medication is encouraging, and it can be deduced that nurses generally adhered to use of the BCMA system when administering doses. However, because the majority of nurses failed to scan enoxaparin boxes, it is not clear whether they were always aware that these could be scanned, or whether they chose not to scan these for practical reasons. As well as the potential patient safety benefits of increasing positive patient identification, we found that for one of 243 scanned doses scanning the medication barcode resulted in the nurse correcting an error that may otherwise have reached the patient.

### 4.2. Comparison to Previous Research

We explored the impact of BCMA as an addition to ePMA rather than a paper-based system and used different measures of workflow, limiting comparison to previous studies [10,11]. Findings differed from one US study where BCMA increased the total medication administration time [12]. However, this was likely due to an increase in time spent on direct patient care during the drug round [12], whereas time spent on medication administration itself did not increase. It is not possible to compare the impact of BCMA on different nursing activity categories as our study did not measure this. The workaround we observed, in which barcode stickers were scanned instead of patients’ wristbands, has also been reported previously [8,9], indicating this may be common to various settings.

### 4.3. Interpretations and Implications for Practice

BCMA appears to make the medication administration process more uniform by requiring nurses to use the medicines cabinet in the COW, thus streamlining the process. It also has potential to reduce time spent on medication documentation by allowing nurses to sign once for all doses rather than individually. Despite this, total drug round time did not differ among our two wards. This may be because a greater number of doses were administered on the BCMA ward, and/or because scanning barcodes did take some time, especially for the patient’s wristband where manoeuvring the COW and then aiming the scanner correctly often appeared difficult. As BCMA is increasingly being introduced within inpatient settings [1], such findings may help with developing more user-friendly designs such as wireless devices that could also be taken into side rooms.

The high rates of active patient identification and medication verification are potentially advantageous to patient safety, although the latter could be improved if a greater proportion of identifiable medications were included in the database, and if nurses better understood which medications could be scanned. This may occur after a longer adjustment period or additional training. Conceivably, BCMA could also lead to complacency as our observations suggest that some nurses believed that they no longer needed to verify details such as expiry dates with the BCMA system.

### 4.4. Strengths and Limitations of Research

As one of few studies to explore the impact of BCMA on nursing workflow, the findings provide a useful basis for further research. A particular strength is that a single observer collected data and made descriptive assessments of workflow, thus supporting consistency of findings. Data collected through observation also overcomes the limitations of self-reporting. 

However, a limitation is that the study took place on two different wards. We had initially hoped to study the same ward before and after BCMA implementation but delays in BCMA implementation on the original study ward meant this was not possible in the time available. While the second ward was very similar to the first, there were differences which may have introduced bias. For instance, the difference in practice around drug round start times has likely affected our data on the timeliness of medication administration, the wards used different COWs, and there was a considerable difference in the number of doses given per drug round and per patient, which will have affected drug round duration. Other differences in practice cannot be ruled out. Data collection on the BCMA ward took place relatively soon after implementation, and staff may not have been fully familiar with the system. Our study is also relatively small and it is not known to what extent the findings may generalise to other institutions with different medication administration practices and BCMA systems. Finally, our findings may have been influenced by the effect of the observer on those observed [20], although this would have been present on both wards.

### 4.5. Implications for Future Research

Subsequent research should be conducted on the same wards to enable direct comparison of workflow before and after BCMA introduction, as well as observing drug rounds at different times of day. Studies collecting data over a longer duration and with larger sample sizes would be advantageous, preferably several months after BCMA implementation to reflect practice after an initial adjustment period. Additional use of a time-motion or work sampling method to detect quantitative differences in time spent on various nursing activities would shed further light on the findings. 

## 5. Conclusions

The findings of this study suggest that BCMA does not adversely affect the length of time taken by nurses to administer medication. While our data suggest that medication administration was less timely with BCMA, this should be interpreted with caution and further research is required, studying the same wards pre- and post-BCMA introduction. Overall, BCMA appears to have led to less variability in how nurses undertake medication administration. The significant increase in active patient identification as well as high proportion of medications verified using the system represent potential benefits to patient safety.

## Figures and Tables

**Figure 1 pharmacy-08-00148-f001:**
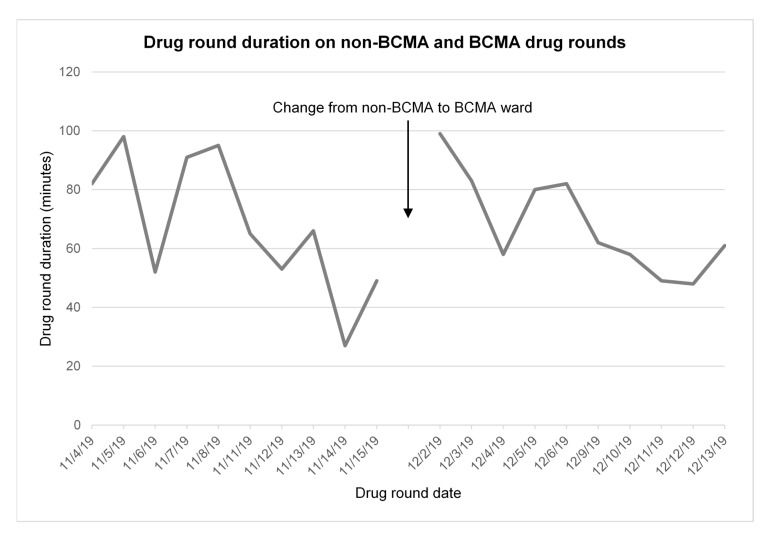
Drug round duration over time on the two study wards. BCMA=Barcode Medication Administration.

**Figure 2 pharmacy-08-00148-f002:**
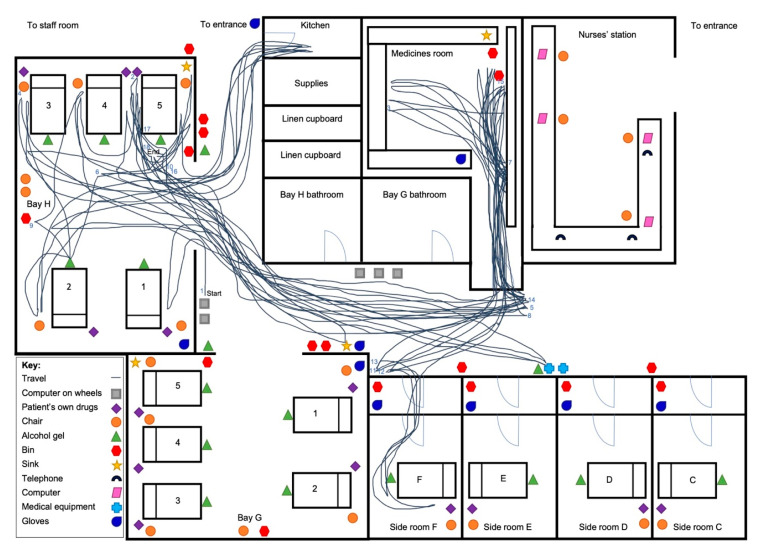
Spaghetti diagram on non-BCMA ward showing nurse walking pattern around the ward. Date: 08/11/2019; Nurse number: 1; Drug round duration: 95 min; Number of patients: 6; Number of doses administered: 23 BCMA = Barcode Medication Administration.

**Figure 3 pharmacy-08-00148-f003:**
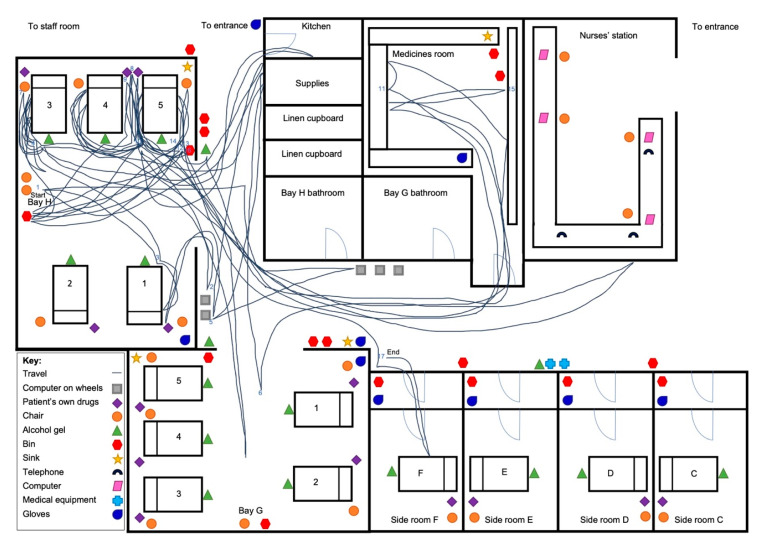
Spaghetti diagram on BCMA ward showing nurse walking pattern around the ward. Date: 06/12/2019; Nurse number: 12; Drug round duration: 82 min; Number of patients: 5; Number of doses administered: 26, BCMA = Barcode medication administration.

**Table 1 pharmacy-08-00148-t001:** Outcome measures with method for data collection. COW = Computer on wheels; BCMA = Barcode medication administration.

Outcome Measure	Method
Drug administration round duration	The time from opening the electronic record on the COW for the first patient to the time when the last medication dose was recorded as being administered on the COW for the last patient was recorded for each drug round. The timed drug round therefore encompassed any activities the nurse performed on the round and any interruptions. The number of patients taking one or more medications, the number of doses that were due and the number of doses that were administered were also recorded to facilitate data analysis.
Timeliness of medication administration	The time for which each dose of medication was scheduled and the time when it was actually administered was recorded. This included only doses of medication administered that could directly be observed as having been given. Medications that were administered ‘when required’ were excluded since a time difference could not be calculated for these. The difference between the administered time and the scheduled dose time was then determined for each dose on both wards.
Active positive patient identification activity	On the non-BCMA ward, an active positive identification was defined in line with hospital policy as the nurse confirming at least two patient identifiers (such as name and date of birth) either by manually checking the patient’s wristband or verbally asking the patient prior to medication administration. On the BCMA ward, an active positive identification was defined as the nurse scanning the barcode on the patient’s wristband or confirming two patient identifiers as before. The method used in each case was also recorded.
Active positive medication dose verification activity	The active verification of doses was only measured on the BCMA ward as it was not possible to determine whether a nurse had verified the medication as part of their thought process without BCMA. A positive verification involved the nurse scanning the barcode on the medication; we recorded whether this was done or not for each dose, together with any reasons why the medication doses were not scanned, and any potential workarounds observed.

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
