# Peer review of "An Evaluation of the Impact of Barcode Patient and Medication Scanning on Nursing Workflow at a UK Teaching Hospital"

_pharmacy, 2020, doi:10.3390/pharmacy8030148_

Round 1

Reviewer 1 Report

General comment

The authors report a small observational study evaluating whether barcode medication administration (BCMA) prolongs the length of time taken by nurses to administer medication. The study was well planned as a prior to an after BCMA introduction study, but was performed on two comparable surgical wards (gastrointestinal vs vascular surgery) with and without BCMA for time reasons.

The methodology, results and also the limitations of the study (short duration, small sample size) are adequately presented and addressed. This pilot study communicates that BCMA does not adversely affect the length of time taken by nurses to administer medication.

Specific comment

Line 154: Scanning a barcode sticker from the patient’s notes instead of the wristband. It should at least be discussed that in BCMA systems wristband scanning is obligatory, otherwise it does not reduced administration of a drug to a wrong patient.  

Line 253: It is well known that participation in a clinical study may produce better performance and results than in daily life. This phenomenon should be mentioned as possible bias, but it should not be mentioned as Hawthone effect. Otherwise the Hawthone effect should be explained. This is not easy. According to a systematic review there is no single, specific Hawthorne effect (McCambridge J, Witton J, Elbournec DR. Systematic review of the Hawthorne effect: New concepts are needed to study research participation effects. J Clin Epidemiol. 2014; 67: 267–277). Consequences of research participation for behaviors being investigated have been found to exist in most studies, although little can be securely known about the conditions under which they operate, their mechanisms of effects, or their magnitudes.

Reviewer 2 Report

Thank you for the submission. It is quite interesting and well-written, although not novel. The key limitations (e.g. different wards) are also mentioned.

Were the data checked appropriately for Normality?  Non-parametric statistical methods are probably more appropriate, given the large variability noted.

The mean number of drug doses administered differed substantially between the wards, which makes the absolute medication round time a very limited comparative measure. The most useful valid outcome is probably the difference in mean (or median?) time per dose - 4.2 minutes (range 3.3 to 5.5 min) on the non-BCMA ward and 2.3 minutes (range 1.8 to 3.2 min) on the BCMA ward (p< 0.001; t-test). While this is mentioned briefly, it is odd that this was not the primary outcome measure (and thereby standardising for the number of doses administered per round).

It would have been useful to also assess nurse satisfaction with the drug administration processes on each ward.
